

# Transcriptome analysis of the oriental melon (*Cucumis melo* L. var. *makuwa*) during fruit development

Ah-Young Shin[1], Yong-Min Kim[2], Namjin Koo[2], Su Min Lee[3], Seokhyeon Nahm[3] and Suk-Yoon Kwon[1,4]

[1] Plant Systems Engineering Research Center, Korea Research Institute of Bioscience and Biotechnology (KRIBB), Daejeon, Korea
[2] Korean Bioinformation Center, Korea Research Institute of Bioscience and Biotechnology (KRIBB), Daejeon, Korea
[3] R&D Devision, Nongwoo Bio Co., Ltd., Yeoju, Kyonggi-do, Korea
[4] Biosystems and Bioengineering Program, University of Science and Technology, Daejeon, Korea

Corresponding author
Suk-Yoon Kwon, sykwon@kribb.re.kr

## ABSTRACT

**Background**. The oriental melon (*Cucumis melo* L. var. *makuwa*) is one of the most important cultivated cucurbits grown widely in Korea, Japan, and northern China. It is cultivated because its fruit has a sweet aromatic flavor and is rich in soluble sugars, organic acids, minerals, and vitamins. In order to elucidate the genetic and molecular basis of the developmental changes that determine size, color, and sugar contents of the fruit, we performed *de novo* transcriptome sequencing to analyze the genes expressed during fruit development.

**Results**. We identified a total of 47,666 of representative loci from 100,875 transcripts and functionally annotated 33,963 of the loci based on orthologs in *Arabidopsis thaliana*. Among those loci, we identified 5,173 differentially expressed genes, which were classified into 14 clusters base on the modulation of their expression patterns. The expression patterns suggested that the differentially expressed genes were related to fruit development and maturation through diverse metabolic pathways. Analyses based on gene set enrichment and the pathways described in the Kyoto Encyclopedia of Genes and Genomes suggested that the expression of genes involved in starch and sucrose metabolism and carotenoid biosynthesis were regulated dynamically during fruit development and subsequent maturation.

**Conclusion**. Our results provide the gene expression patterns related to different stages of fruit development and maturation in the oriental melon. The expression patterns give clues about important regulatory mechanisms, especially those involving starch, sugar, and carotenoid biosynthesis, in the development of the oriental melon fruit.

## INTRODUCTION

The oriental melon (*Cucumis melo* L. var. *makuwa*) is one of the most important diploid crops within the Cucurbitaceae family, presenting highly variable fruit traits, such as flesh color, sugar contents, and shape, among different cultivars. They are grown primarily
for their fruit, which generally have a sweet aromatic flavor and contains soluble sugars, organic acids, minerals, and vitamins (*Burger et al., 2006*; *Fernández-Trujillo et al., 2011*; *Nunez-Palenius et al., 2008*).

Breeding programs for the oriental melon have mainly focused on the selection of traits associated with fruit appearance and textural attributes. Among the different parts of the melon plant, the fruit has the highest diversity in size (from 4 cm up to 200 cm), shape (round, oval, flat, obovoid, or long), and internal and external colors (orange, green, white, pink, or even mixtures of those colors). Variation is also expressed in the rind color and rind texture (*Monforte et al., 2014*; *Nunez-Palenius et al., 2008*). The developmental processes of the fruit can be divided into three phases. The initial phase involves the floral meristem formation and flower development. In the second phase, the fruit grows primarily through cell division. The third phase is characterized by cell expansion (*Gillaspy, Ben-David & Gruissem, 1993*; *Higashi, Hosoya & Ezura, 1999*). Once the fruit reaches its full size, the ripening process is initiated, highlighted by major biochemical changes in the maturing fruit. Depending on the plant species, the ripening process is typically associated with dramatic changes in color, aroma, and fruit structure (*Gapper, McQuinn & Giovannoni, 2013*; *Giovannoni, 2004*; *Monforte et al., 2014*).

Sucrose, glucose, and fructose are the major soluble sugars, and sucrose is the predominant sugar in the melon at maturity. High levels of sucrose convey sweetness to the melon fruit (*Burger et al., 2003*; *Villanueva et al., 2004*). A previous study showed that the patterns of sugar accumulation during fruit maturation differed between the oriental melon and other melons (*Saladie et al., 2015*). The levels of glucose and fructose were similar among one oriental melon (Korean landrace Sunghwan, PI161375) and three other melons (Vedrantais, Dulce, and Piel de sapo). The levels of sucrose were very low early in fruit development and subsequently increased starting 30 days after pollination (DAP; Vedrantais, Piel de sapo, and PI161375) or 40 DAP (Dulce). Later, the sucrose levels dropped in the oriental melon (PI161375) but reached a plateau in the other melons. Genes linked to sugar metabolism were differentially expressed between the oriental melon and the other melons. These results indicated that differences in gene expression related to sugar metabolism might determine the sucrose contents of the fruits. Thus, the sugar contents or sugar sink in the fruit of the oriental melon could also differ from those of other melons.

With increasing interest in its biological properties and economic importance, the oriental melon has become an attractive model for studying valuable features, such as fruit morphology (*Saladie et al., 2015*), tissue specific RNA-seq (*Cai, Wang & Pang, 2015*; *Cao et al., 2016*; *Kim et al., 2016b*; *Liu et al., 2016*), pathogen resistance (*Bezirganoglu et al., 2013*), and understand the specialization of some genes involved in plant growth (*Cai, Wang & Pang, 2015*; *Cao et al., 2016*; *Kim et al., 2016b*; *Liu et al., 2016*). Despite these studies, the molecular mechanisms of fruit development in the oriental melon are still not elucidated.

Here, we report a *de novo* transcriptome assembly and analysis of the oriental melon, KM (Korean landrace; Gotgam), during fruit development. Transcriptome and Kyoto Encyclopedia of Genes and Genomes (KEGG) pathway analyses revealed enrichment of differentially expressed genes (DEGs) in eight pathways, especially carotenoid biosynthesis, starch and sucrose metabolism. These results provide information about the genes involved

in fruit development in the oriental melon and will help elucidate the basis of fruit maturation and sugar contents in that economically important fruit.

## MATERIALS AND METHODS

### Plant materials

Oriental melon plants (Gotgam) were grown in a greenhouse with plastic pots containing a soil mixture. The temperature was maintained at 28 °C during the day and 24 °C during the night Supplemental lights were used to provide an 18-h light period. Flowers were hand pollinated and at least three fruits were harvested from different plants at 8, 16, 24 and 32 DAP. Tissues were immediately frozen in liquid nitrogen and stored at −80 °C until analysis.

### Total RNA extraction, *de novo* assembly and annotation

Total RNA was isolated from each sample of fruit flesh using TRIzol reagent according to the manufacturer's instructions. Purified total RNAs were used to synthesize cDNAs, which were amplified according to the Illumina RNA-Seq protocol and sequenced using an Illumina HiSeq2000 system, producing 17.9 Gb 101-bp paired-end reads in total from one sequencing run with pooled by three independent fruits in each stage. The sequence data were filtered by quality score ($Q \geq 20$) and minimum length ($\geq 25$ bp) using SolexaQA (*Cox, Peterson & Biggs, 2010*). The quality-checked sequence reads from different tissue samples were assembled *de novo* using two software tools based on de Bruijn graphs: Velvet (v1.2.07) (*Zerbino & Birney, 2008*) and Oases (v0.2.08) (*Schulz et al., 2012*). The assembled transcripts were used for DEG extraction or functional annotation by edgeR or BLASTP with *Cucumis melo* or *Arabidopsis thaliana* as queries (*Kim et al., 2015*; *Klukas & Schreiber, 2007*). The annotated genes were used for further analyses including gene ontology (GO) and KEGG analyses.

### Differentially expressed genes and clustering analysis

DEGs were selected by comparing two distinct stages of fruit development. The reads with each sequence tag were mapped to the assembled loci using Bowtie (mismatch $\leq 2$ bp); the number of cleanly mapped reads for each locus was determined; and the data were normalized using the DESeq library in R (*Anders & Huber, 2010*). The final DEGs were identified by a change in read coverage of at least twofold, a false discovery rate (FDR) $\leq 0.05$, and a read count $\geq 100$ in both samples. The clustering of the DEGs was performed using the Mfuzz program (*Futschik & Carlisle, 2005*; *Kumar & Futschik, 2007*).

### Functional enrichment and gene ontology analyses

The assembled loci were annotated to the GO database (downloaded from http://www.geneontology.org/) using BLASTP (e-value $= 1e^{-06}$). The annotation was carried out using the GO classification results from the Map2Slim.pl script (*Kim et al., 2016a*). Functional enrichment analysis was carried out using DAVID, a web-accessible program that provides a comprehensive set of functional annotation tools to infer biological meaning from large lists of genes (*Huang Da, Sherman & Lempicki, 2009*; *Huang, Sherman & Lempicki, 2008*). Fisher's exact test was used to analyze the gene lists annotated with the TAIR identifications

A

B

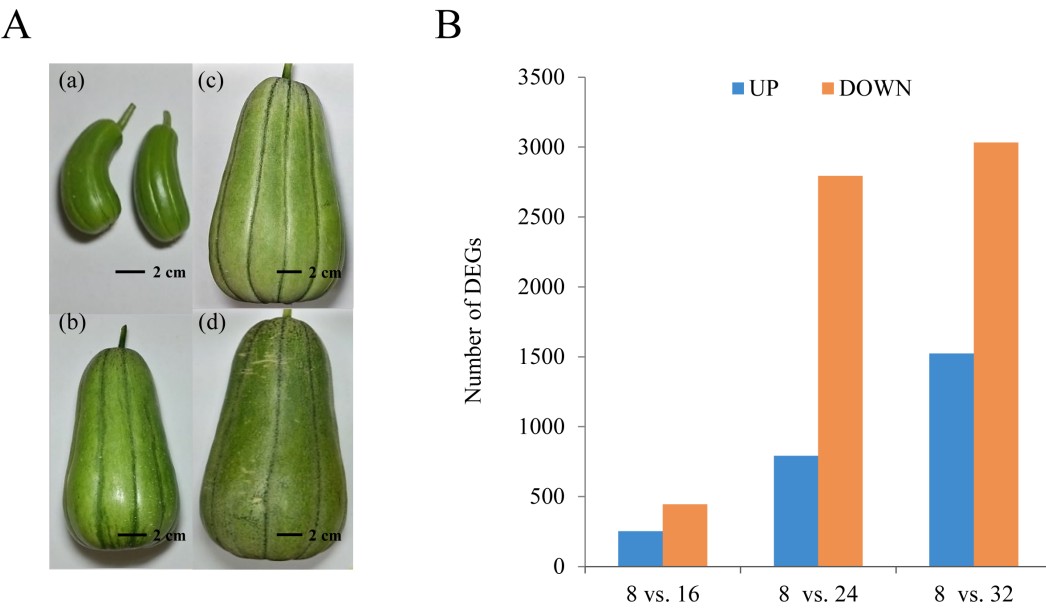

**Figure 1** **Growth and development of oriental melon fruits.** (A) Developmental changes of KM fruit during growth and maturation. Fruits are shown 8 (a), 16 (b), 24 (c), and 32 (d) days after pollination (DAP). (B) Number of differentially expressed genes in each comparison: 8 DAP vs. 16 DAP, 8 DAP vs. 24 DAP, and 8 DAP vs. 32 DAP. 'A' was the control, and 'B' was the experimental group in 'A vs. B.'

of the transcripts with respect to the GO terms under the following criteria: count $\geq$ 10; FDR $\leq$ 0.05.

## Kyoto Encyclopedia of Genes and Genomes analysis

KEGG analysis was performed to investigate the biochemical pathways involved in fruit development using the DEGs among the KM transcripts. *A. thaliana* orthologs of the KM DEGs in each cluster were identified, and KEGG analysis was performed with the Plant GSEA platform (http://structuralbiology.cau.edu.cn/PlantGSEA/analysis.php) (*Yi, Du & Su, 2013*) using the default parameters.

## RESULTS

### *De novo* sequence assembly and functional annotation

The KM fruit has a rounded, rectangular shape, medium size, deep-green skin, and light-red flesh. The progression of growth and development in the KM fruit is highly reproducible and includes visible external and internal morphological changes. The fruit was still small at 8 DAP. From 8 DAP to 16 DAP, the diameter and length of the fruit increased rapidly. After 16 DAP, the fruit volume continuously increased to the maximum, and the color changed to dark green (Fig. 1A).

For transcriptome analysis of the fruit development and maturation, we collected samples at two developmental stages (8 DAP and 16 DAP) and two maturation stages (24 DAP and 32 DAP) for total RNA extraction. Sequencing with Illumina HiSeq 2000 produced 17.9 Gb of paired-end data in total. To determine a unigene set, we performed *de novo* transcriptome assembly with Velvet/Oasis and assembled 100,875 transcripts. In general, the assembled

**Table 1** Statistics of the assembled transcripts from oriental melon.

| Length | KM_locus | KM_transcripts |
|---|---|---|
| ~500 | 24,747 | 29,299 |
| 501–1,000 | 8,045 | 17,829 |
| 1,001–1,500 | 3,966 | 14,826 |
| 1,501–2,000 | 3,376 | 12,980 |
| 2,001–2,500 | 2,590 | 9,191 |
| 2,501–3,000 | 1,743 | 6,213 |
| 3,001–3,500 | 1,254 | 4,152 |
| 3,501–4,000 | 764 | 2,537 |
| 4,001–4,500 | 487 | 1,600 |
| 4,501~ | 694 | 2,248 |
| Total | 47,666 | 100,875 |

transcripts contain the alternative spliced form of transcripts or partial assembled transcripts. To remove these transcripts and determine representative loci, we performed reciprocal blast and the longest and non-redundant transcripts were determined. Then, BLASTx was performed with remained transcripts using Non-Redundant (NR) database and 1,722 of transposable element related genes were removed. Finally, 47,666 transcripts were remained as representative loci (Table 1 and Fig. S1). We then annotated the representative loci by BLASTP (e-value $\leq$ 1e$^{-08}$) against melon (CM V3.5) and Arabidopsis genes (TAIR 10), as reported previously (*Kim et al., 2016a*). Furthermore, we also performed ISGAP pipeline to determine more precise transcripts by removing intron contaminants (*Kim et al., 2015*). Of the representative loci, 35,364 were assigned to melon reference proteins, and 33,963 were assigned to Arabidopsis proteins. For further analyses, we used the representative loci, including the functionally annotated loci, to remove alternative transcripts and assembly errors.

## Gene ontology classification of differentially expressed genes

To examine differential gene expression, we mapped the trimmed raw reads to the representative loci to investigate KM specific transcripts and their expression patterns and determined the genes that were expressed differentially between the indicated time points (Fig. 1B). The results showed that 253 and 446 genes were up-regulated and down-regulated, respectively, at 16 DAP relative to 8 DAP. In addition, at 24 DAP and 32 DAP, 792 and 2,794 genes were up-regulated, respectively, while 1,523 and 3,034 genes were down-regulated, respectively. These results showed that many genes were transcriptionally regulated during fruit maturation and that various metabolic pathways involved in fruit development or maturation were dynamically regulated at the transcriptional level. To investigate the functions of the DEGs in each developmental stage, we performed a GO analysis to classify the DEGs (Fig. 2 and Fig. S2). We assigned a total of 4,557 DEGs to 38 functional categories: 19 'biological processes,' 11 'cellular components,' and eight 'molecular functions' (Fig. 2). In each of those categories, 'cellular process' (7.75%), 'cell and cell part' (20.84%), and 'binding and catalytic activity' (10.60%) were the most abundant GO terms, respectively.

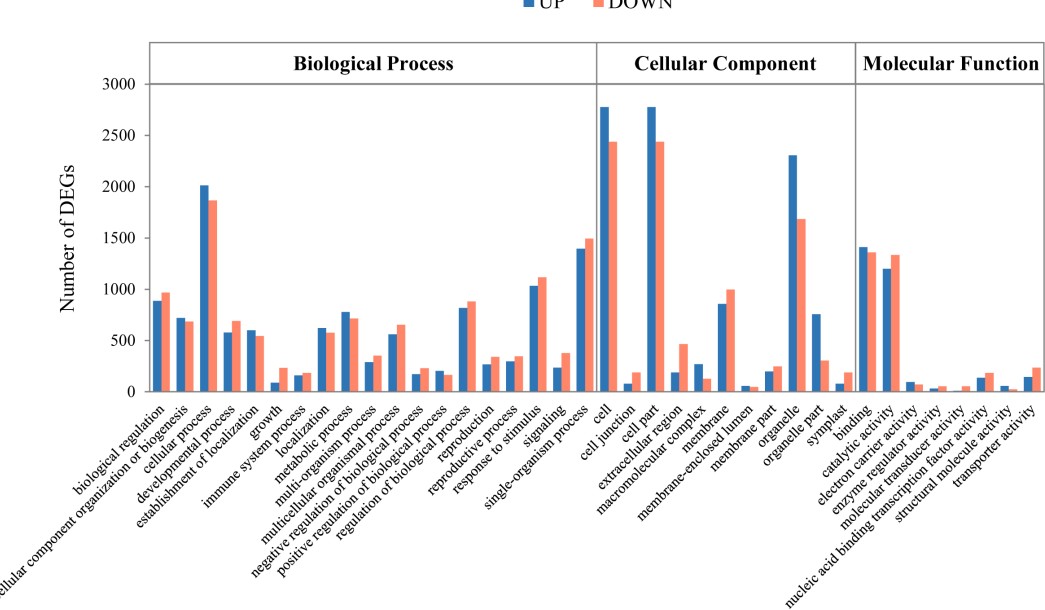

**Figure 2** **Functional categorization of the genes with significant transcriptional differences between developing and mature fruits.** Differentially expressed genes (DEGs; 8 days after pollination (DAP) vs. 32 DAP) were classified into three functional categories: 'Biological process', 'Cellular component', and 'Molecular function.' 'A' was the control, and 'B' was the experimental group in 'A vs. B.' The *x*-axis indicates the subgroups in the Gene Ontology annotation, while the *y*-axis indicates the number of DEGs in each main category.

The GO analysis revealed that both the up-regulated gene set and the down-regulated gene set were significantly enriched with genes involved in 'catalytic activity' and 'binding activity'. Those results suggest that genes involved in catalytic pathways were regulated dynamically during fruit maturation.

## Clustering and Kyoto Encyclopedia of Genes and Genomes pathway enrichment of the differentially expressed genes

To better understand the transcriptional regulation of the gene expression involved in fruit development and maturation, we performed a cluster analysis of the DEGs (*Futschik & Carlisle, 2005*; *Kumar & Futschik, 2007*). We classified a total of 5,173 DEGs into 14 types of clusters based on the modulation of expression patterns (Fig. 3 and Table 2).

To investigate the biological functions of the candidate genes potentially involved in fruit development and maturation, we used the 5,173 DEGs for further KEGG analysis (*Kanehisa et al., 2012*). We identified orthologs of the KM loci in *A. thaliana* and applied the orthologs for KEGG analysis. The KEGG analysis identified eight pathways including starch and sucrose metabolism (ath00500), carotenoid biosynthesis (ath00906), phenylpropanoid biosynthesis (ath00940), and pentose and glucoronate interconversions (ath00040) (Table 3). Those results indicated that genes involved in metabolic pathways were transcriptionally regulated during fruit development (8 DAP and 16 DAP) and maturation (24 DAP and 32 DAP). We mapped a total of 48 DEGs to starch and sucrose metabolism pathways and

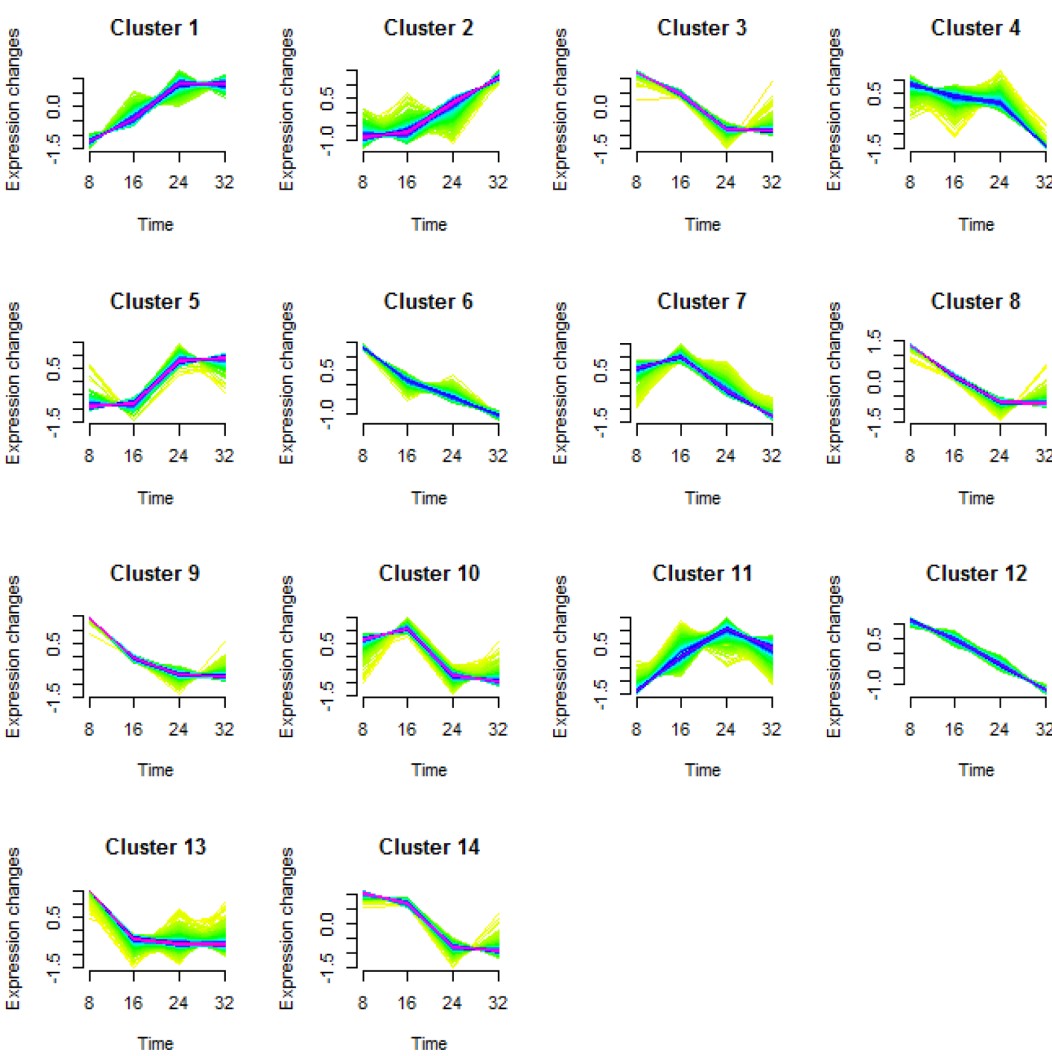

**Figure 3 Cluster analysis of the differentially expressed genes in each comparison.** Differentially expressed genes (DEGs) in KM were categorized into 14 clusters depending on their expression during fruit growth and maturation. The levels of gene expression are represented along the *y*-axis as fold changes, and the stages of fruit maturation are represented along the *x*-axis as 8, 16, 24, and 32 days after pollination (DAP).

grouped them into 11 clusters (Fig. 4). Among various pathways, glucose metabolic pathways showed finely tuned expression patterns of genes. Two glucose metabolic pathways showed high gene expression patterns in both fruit development and maturation stages (red circles in Fig. 4A) and one pathways showed high gene expression patterns in maturation stage (orange circle in Fig. 4A). Six loci (Locus_27541, Locus_1366, Locus_1187, Locus_7852, Locus_30074, and Locus_8322) were involved in three glucose-metabolism pathways with different substrates such as UDP-glucose, cellulose, and maltose (Fig. 4A). These loci were highly expressed during fruit development and subsequently down-regulated during fruit maturation. In contrast, three loci (Locus_7857, Locus 12354, and Locus_5309) showed increased expression during fruit maturation. Those results indicated that nine

**Table 2   Numbers of genes in each DEG cluster.**

| Cluster No. | Number of DEGs |
|---|---|
| Cluster 1 | 605 |
| Cluster 2 | 286 |
| Cluster 3 | 414 |
| Cluster 4 | 325 |
| Cluster 5 | 613 |
| Cluster 6 | 636 |
| Cluster 7 | 319 |
| Cluster 8 | 322 |
| Cluster 9 | 272 |
| Cluster 10 | 190 |
| Cluster 11 | 234 |
| Cluster 12 | 235 |
| Cluster 13 | 188 |
| Cluster 14 | 534 |
| Total | 5,173 |

**Table 3   Enriched pathways from the KEGG analysis.**

| ID | Description | Count | p. adjust | q value |
|---|---|---|---|---|
| ath00940 | Phenylpropanoid biosynthesis | 43 | 0.000136355 | 0.000114826 |
| ath00196 | Photosynthesis-antenna proteins | 12 | 0.000193075 | 0.000162589 |
| ath04075 | Plant hormone signal transduction | 62 | 0.000193075 | 0.000162589 |
| ath00500 | Starch and sucrose metabolism | 48 | 0.000614771 | 0.000517701 |
| ath00906 | Carotenoid biosynthesis | 13 | 0.000638312 | 0.000537526 |
| ath00592 | Alpha-Linolenic acid metabolism | 14 | 0.001779272 | 0.001498335 |
| ath00195 | Photosynthesis | 22 | 0.004099069 | 0.003451848 |
| ath00040 | Pentose and glucuronate interconversions | 22 | 0.007804406 | 0.006572132 |

genes in three glucose-metabolism pathways were involved in the regulation of glucose contents in the oriental melon fruit. Genes involved in fructose metabolism were highly expressed during fruit development (8 DAP) and subsequently down-regulated during fruit maturation (purple circle in Fig. 4A). Furthermore, genes involved in UDP-glucose and sucrose-6P metabolism, both precursors of sucrose, were up-regulated at 8 DAP and down-regulated during fruit maturation (purple circles in Fig. 4A).

The KEGG analysis identified 13 genes involved in carotenoid biosynthesis. In the carotenoid-biosynthesis pathway, phytoene synthase (PSY, EC 2.5.1.32) (*Bouvier, Rahier & Camara, 2005*), an enzyme that catalyzes the synthesis of 40-carbon phytoene from the condensation of two geranylgeranyl diphosphate (GGPP) molecules (*Bouvier et al., 2005*), was up-regulated during fruit maturation. This step is the first step in carotenoid biosynthesis. In contrast to PSY, three carotenoid hydroxylases (CYP97A3) (*Kim & DellaPenna, 2006*) were up-regulated during fruit development and subsequently down-regulated during fruit maturation. For xanthophyll biosynthesis, three 9-cis-epoxycarotenoid dioxygenases

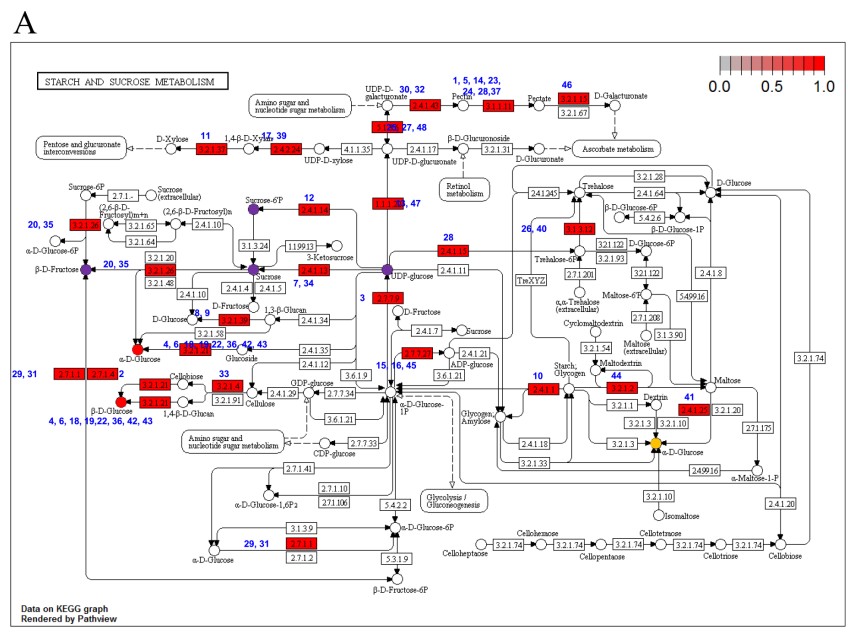

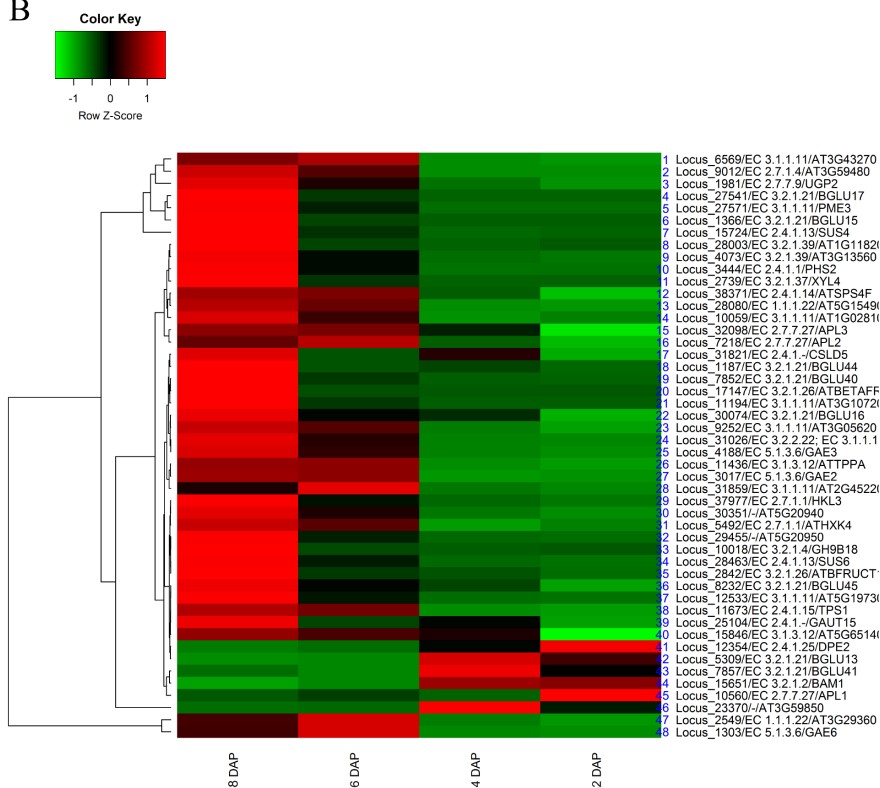

**Figure 4  Pathway and differential expression analysis of starch and sucrose metabolism.** (A) Starch and sucrose metabolism diagram from the KEGG Pathway analysis. Color-coded enzymes indicate target genes with differential expression. The numbers depicted in color-coded enzymes indicate genes in heat map. 

**Figure 4 (…continued)**
The purple circles of glucose metabolic pathways indicate high expression patters in fruit developmental stage and orange circle indicates high expression patterns in fruit mature stage. The red circles indicate high gene expression patterns in both fruit developmental and mature stages. (B) Differentially expressed genes (DEGs) from the starch and sucrose metabolism diagram. The locus tag and enzyme code shown in starch and sucrose metabolism are described in a heat map.

(*Schwartz et al., 1997*) (NCEDs), which yield a C15 intermediate xanthoxin, were differentially expressed during fruit development and maturation. NCED5 was up-regulated during fruit development, whereas NCED3 and NCDE4 were up-regulated during fruit maturation. In addition, three CYP707 genes (*Millar et al., 2006*) involved in ABA oxidation were up-regulated during fruit development (Fig. 5). In mature melon fruits, beta-carotene is the most abundant carotenoid, with trace amounts detected even in the white-fleshed Piel de sapo (*Saladie et al., 2015*). Our results corresponded to previous results and indicated that the expression patterns of carotenoid biosynthesis genes were regulated to increase the carotenoid contents during fruit maturation.

## DISCUSSION

We identified 47,666 genes expressed in the KM oriental melon fruit during development. Among those, we identified 5,173 DEGs and assigned them to various functional categories: 19 'biological processes, 11 'cellular components, and eight 'molecular functions'. Hierarchical clustering analysis grouped the DEGs into 14 types of clusters based on the modulation of their expression patterns. The candidate genes were assigned to eight KEGG pathways including starch and sucrose metabolism (ath00500), carotenoid biosynthesis (ath00906), phenylpropanoid biosynthesis (ath00940), and pentose and glucoronate interconversions (ath00040).

In a previous study, glucose and fructose were found to be synthesized early (5 DAP) and to be maintained at similar levels during fruit maturation. In melons, the sugar content is very low early in fruit development and subsequently increases around 30 DAP (Vedrantais, and Piel de sapo) or 40 DAP (Dulce). The sucrose content reaches a plateau in many melons. The sugar accumulation patterns of the oriental melon (PI161375; Sunghwan) were found to be similar to those of other melons except for a clear drop in the sucrose content during maturation (*Saladie et al., 2015*). This research suggested that the regulation of sugar metabolism differs between oriental melons and other types of melons. The KEGG pathway results and DEG analyses in our study suggest that genes involved in the glucose and fructose metabolic pathways were up-regulated early during fruit development (8 DAP), which is consistent with the previous results. The genes involved in sucrose synthesis were activated early in fruit development and subsequently down-regulated from 16 DAP to 32 DAP (Fig. 4). This result suggests that sucrose is synthesized early in fruit development and used as a precursor for fructose or glucose. In a previous study, the level of sucrose was elevated during fruit maturation, and orthologs of sucrose synthase were detected, suggesting that multiple sucrose synthases were involved in sucrose synthesis during fruit maturation (*Saladie et al., 2015*).

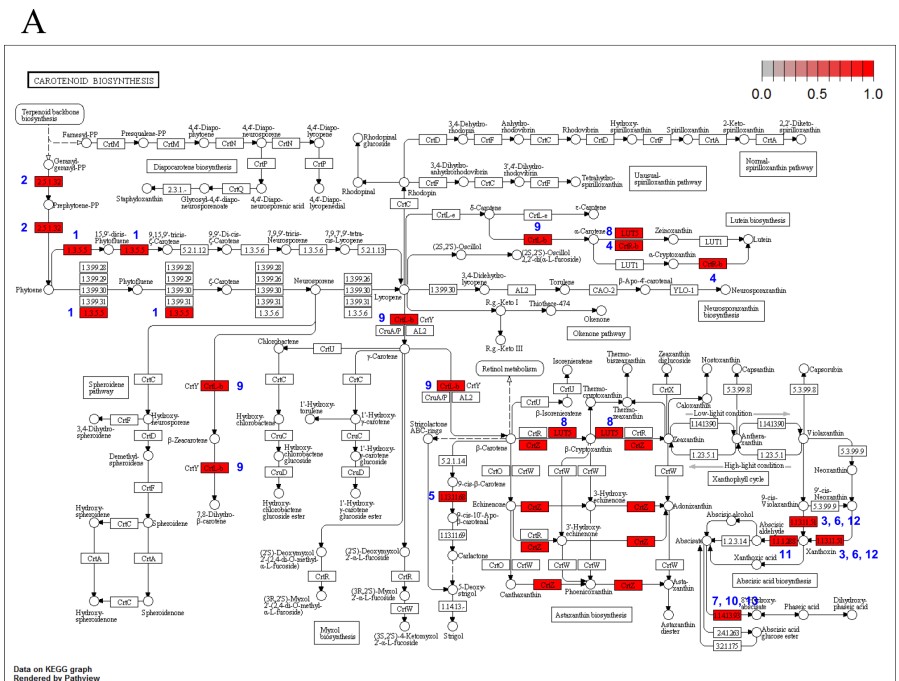

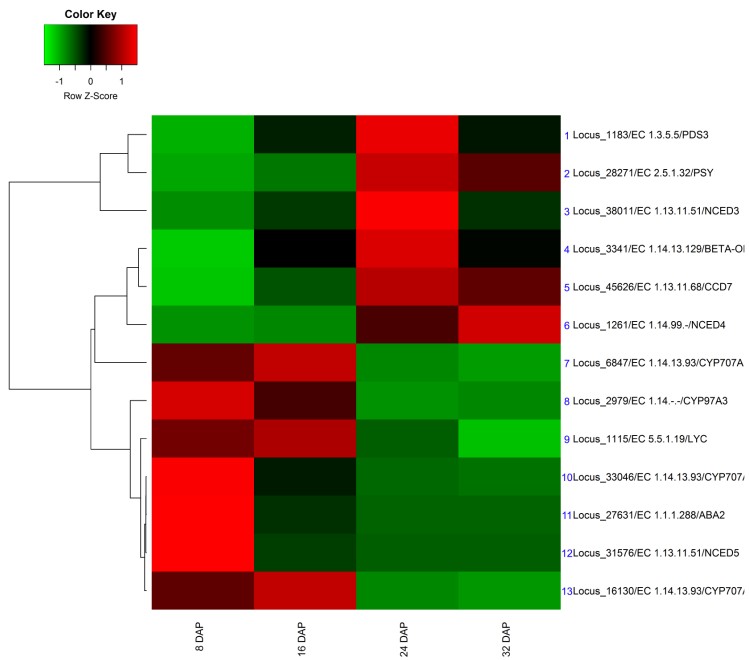

**Figure 5** **Pathway and differential expression analysis of carotenoid-biosynthesis pathway.** (A) Carotenoid-biosynthesis diagram from the KEGG Pathway analysis. Color-coded enzymes indicate target genes with differential expression. The numbers depicted in color-coded enzymes indicate genes in heat map. (B) Differentially expressed genes (DEGs) from the carotenoid-biosynthesis pathway diagram. The locus tag and enzyme code shown in carotenoid biosynthesis are described in a heat map.

Carotenoids, a group of isoprenoid molecules generally regarded as pigments, provide protection against photo-oxidative damage (*Ruiz-Sola & Rodríguez-Concepción, 2012*). Carotenoids are grouped into two major classes: carotenes and xanthophylls. Among the xanthophylls, xanthoxin is a precursor of the plant hormone ABA. Our analyses revealed that the expression of genes involved in the biosynthesis of carotenes and ABA precursors were up-regulated during fruit maturation (Fig. 5). In addition, high expression levels of CYP707 genes (*Millar et al., 2006*) and the expression patterns of NCED genes (*Schwartz et al., 1997*) suggested that gene expression involved in ABA biosynthesis or inactivation led to low ABA content during fruit development and high ABA content during fruit maturation. These results suggest that the expression of carotenoid biosynthesis genes was finely tuned and dynamically regulated through the use of various orthologs.

The results of the KEGG analysis suggested that the proportion of fruit flesh in the samples might be higher than that of other tissues. This feature might have affected the observed gene expression patterns. Precise analyses of the metabolic pathways during fruit maturation were difficult to perform using transcriptomes derived from pooled samples containing pericarp, flesh, and placentae. Tissue-specific transcriptome analyses will be required to further unveil the metabolic pathways involved in fruit maturation and important properties such as the sugar and carotenoid contents. Our results provide clues about the transcriptional regulation of metabolic genes involved in fruit development and maturation in the oriental melon. Further development-specific and tissue-specific transcriptome analyses comparing oriental melons and other types of melons will provide more understanding of the transcriptional regulation of metabolic genes during fruit development and maturation.

In conclusion, transcriptome and pathway analyses of the oriental melon suggest that three glucose-metabolism pathways are involved in the determination of glucose content, and the genes involved in those pathways are regulated dynamically during fruit maturation.

### Funding
This work was supported by a grant from the Biogreen21 program (PJ011088) of the Rural Development Administration, Korea, and the KRIBB Research Initiative Program. The funders had no role in study design, data collection and analysis, decision to publish, or preparation of the manuscript.

### Grant Disclosures
The following grant information was disclosed by the authors:
Biogreen21 program of the Rural Development Administration, Korea: PJ011088.
KRIBB Research Initiative Program.

### Competing Interests
Su Min Lee and Seokhyeon Nahm are employees of Nongwoo Bio Co., Ltd., Yeoju, Kyonggi-do, Korea.

## Author Contributions

- Ah-Young Shin conceived and designed the experiments, performed the experiments, analyzed the data, wrote the paper, prepared figures and/or tables.
- Yong-Min Kim conceived and designed the experiments, analyzed the data, wrote the paper, prepared figures and/or tables.
- Namjin Koo, Su Min Lee and Suk-Yoon Kwon contributed reagents/materials/analysis tools.
- Suk-Yoon Kwon conceived and designed the experiments, wrote the paper, reviewed drafts of the paper.

## Data Availability

GenBank: SRX2181402 (PRJNA300582).

## Supplemental Information

Supplemental information for this article can be found online at http://dx.doi.org/10.7717/peerj.2834#supplemental-information.

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
