# Peer review of "Transcriptome analysis of the oriental melon (Cucumis melo L. var. makuwa) during fruit development"

_PeerJ, doi:10.7717/peerj.2834_

## Round 0.1 · original submission · Minor Revisions

When you submit the revised version, please reply point by point to the reviewers comments.

Reviewer 1 ·

Basic reporting

The oriental melon is one of the economically important crops within the Cucurbitaceae which ranks among the highest of plant families used as human food. The majority of breeding strategy for the oriental melon is the selection of fruit-related traits such as color, texture, and sweetness. The authors reports that transcriptomic anlaysis of the oriental melon used as the useful resource for disease resistance showed dynamic gene expression change in variable metabolic pathways related fruit development. I believe that this manuscript is very useful and important for further analysis of fruit-related traits and breeding programs. But it needs to correct/edit some point and describe more details to clear up

Experimental design

Fig.4 and Fig5, It need to improve the quality of figures. For E.g. suggesting that the size of color key should be manipulated to proper.

Page 6 line 8 and 19, describe DAP and DEG as full word, next time it can be used as an abbreviation. Throughout manuscript plz check abbreviations before use it whether full words are used or not

I wonder that any reference to build up the strategy for de novo assembly and/or DEG analysis

Page 7 line 13, plz, cite the ref. on Map2Slim.pl script

Validity of the findings

Page 8 line 20, plz indicate the sentence more clearly, what are the representative loci and functionally annotated loci ? Those are derived from de novo assembly or assigned to melon gene model of melon, respectively?

There are informatics figures and tables, which are no doubt to be useful for audience. But there are no detailed information. I’d like to suggest deposit the supplementary table(s) including de novo assembled transcript seq, their matched GO analysis info. and clustered DEG info and so on. I could be more helpful and useful clues to understand fruit-related traits in melon and others for further studies.

Additional comments

Page 4 line 7-9, the sentence should be clear for understanding the meaning.
Page 4 line 22, the meaning of the sentence, “the oriental melon differ from other melons in the importance of fruit sweetness in its breeding” is ambiguous. How about Move the sentence to proper region in the paragraph. E.g. page 5 line 2 ahead of the sentence “A previous study~”
Page 5 line 13-15, the sentence, “Recently, several studies of oriental melons have been reported~” should be explained more clearly for what.
Page 5 in third paragraph, the contents need to be precisely and connotatively.

Reviewer 2 ·

Basic reporting

No comments

Experimental design

No comments

Validity of the findings

No comments

Additional comments

The manuscript by Shin et al., presents transcriptome analyses of Cucumis melo L.. The authors provide transcriptional profiles and differentially expressed gene information during fruit development and maturation. Especially, the authors suggest that the identified DEGs in this study affect regulation of pathways involving starch and sucrose metabolism and carotenoid biosynthesis. I thought that the generated data and biological implication will be useful resources for study in field of Cucumis genomics and transcriptomics. Therefore, I suggest publication of this study in PeerJ journal after modification and resolving of minor concerns.

1. Number of genes (49,338) in the representative transcripts is too many considering annotation in transcriptome without alternative splicing variant. Is it due to copy number increase in Cucumis species or over-estimation derived from assembly? Filtration or detail description is required

2. For users and readers studying Cucumis genomics, the authors need to release expression information of genes and sequences of assembled transcripts as well as genes.

Reviewer 3 ·

Basic reporting

no comments

Experimental design

no comments

Validity of the findings

no comments

Additional comments

In the manuscript entitled “Transcriptome analysis of the oriental melon (Cucumis melo L. var. makuwa) during fruit development (#13416)” the authors conducted de novo transcriptome analysis to characterize fruit development in the oriental melon cultivar ‘KM (Gotgam)’. The purpose, strategy, and methodology of this study are well described in the manuscript. The results also appear to be fine. However, there are some ambiguous points in the manuscript (see comments below). Because these are critical to judge this study, please revise manuscript according to the comments below.

1) The authors used four fruit samples (8, 16, 24, and 32 DAP) in this study. However, I couldn’t understand how many Hiseq sequencing authors conducted in each fruit sample (duplicate or triplicate or just one run?). Please describe exact information in the manuscript.
2) In line 104 in page 9, authors simply described “Purified total RNAs were used to synthesize cDNAs, which were … sequenced using an Illumina HiSeq2000 system, producing 4.9 Gb 101-bp paired-end reads.” However, I couldn’t understand how many Hiseq sequence data they used for de novo assembling. Did authors use just 4.9 Gb in total or 4.9 Gb x 4 samples = 19.6 Gb or 4.9 Gb x 4 samples x triplicate = 58.8 Gb? Please describe exact information in the manuscript.
3) Did authors combine all Hiseq data of four fruit samples to conduct de novo assembly? Or separately conduct it in each sample? (e.g. 8, 16, 24, and 32 DAP).
4) Use of de novo assembly strategy is very excellent. However, I just wondered how many degrees the authors successfully restored connected contings (original mRNA sequences). They described Velvet/Oasis analysis resulted in 100,875 transcripts. But, I think this number is too high. In usual, de novo assemble with Illumina’s Hiseq technology is almost impossible to recover full-length mRNA sequence because read length is too short (around 100 bp). Even if huge amount of Hiseq data (~100 Gb) is used for de novo assemble, it often results in fragmented contigs due to the presence of repeat sequence in mRNA or similar sequences among homologues. If the amount of initial data input is too low, the degree of contig fragmentation becomes much high. Although the authors presented some statistics of assembly (e.g. Table 1), additional information is required to show the connectivity of contigs obtained in this study.
5) To simply identify differentially expressed genes (DEGs), Tophat-cufflinks analysis (reference mapping) seems more suitable. The DHL92 melon genome reference is also available for this purpose. What was the advantage of using de novo assemble strategy compared with reference mapping? Were there any new transcripts that are only present in ‘KM’? Please add discussion or result about this.
6) In page 22, the title of Table 1 is “Statistics of the assembled transcripts from two oriental melons”. What does the “two melon” mean?

---

## Round 0.2 · accepted · Accept

The revised version mostly resolved the concerns raised by reviewers and is acceptable in its current form.